# Nutritional Values of Minikiwi Fruit (*Actinidia arguta*) after Storage: Comparison between DCA New Technology and ULO and CA

**DOI:** 10.3390/molecules27134313

**Published:** 2022-07-05

**Authors:** Tomasz Krupa, Kamila Klimek, Ewa Zaraś-Januszkiewicz

**Affiliations:** 1Department of Pomology and Horticulture Economics, Institute of Horticultural Sciences, Warsaw University of Life Sciences (SGGW-WULS), 159C Nowoursynowska Street, 02-787 Warsaw, Poland; 2Department of Applied Mathematics and Computer Science, University of Life Sciences in Lublin, 20-033 Lublin, Poland; kamila.klimek@up.lublin.pl; 3Department of Environment Protection and Dendrology, Institute of Horticultural Sciences, Warsaw University of Life Sciences (SGGW-WULS), 159C Nowoursynowska Street, 02-787 Warsaw, Poland; ewa_zaras_januszkiewicz@sggw.edu.pl

**Keywords:** hardy kiwi, storage, controlled atmosphere, ultra-low oxygen, polyphenols, flavonols, sugar, organic acid

## Abstract

The dietary properties of minikiwi make them, along with other fruits and vegetables, suitable as the basis for many slimming and pro-health diets. Prolonging the availability of minikiwi can be provided by different storage technologies. This experiment focused on evaluating the effect of various O_2_ and CO_2_ concentrations, i.e., low-oxygen atmosphere (DCA, 0.4% CO_2_:0.4% O_2_; ULO, 1.5% CO_2_:1.5% O_2_) or high-CO_2_ (CA, 5% CO_2_:1.5% O_2_) storage, in order to provide the consumer with fruits with comparable high nutritional values. Evaluation gave the basic characteristics of the fruits that characterize their health-promoting properties, i.e., total polyphenols (TPC), phenolic acids and flavonols, antioxidant activity (AA), monosaccharides, and acid content. The atmosphere with a higher CO_2_ content of 5% (CA) effectively influenced the high value of ascorbic acid even after 12 weeks of storage. DCA technology contributed to a significant inhibition of phenol loss but not as effectively as CA technology. In contrast, glucose and fructose contents were found to be significantly higher after storage in ULO or DCA, while sucrose content was more stable in fruit stored in CA or DCA. CA technology conditions stabilized the citric acid content of minikiwi, while DCA technology was less effective in inhibiting acid loss. The nutritional value of the fruit after storage in CA or DCA was not significantly reduced, which will allow the supply of fresh minikiwi fruit to be extended and provide a valuable component of the human diet.

## 1. Introduction

Due to increasing consumer awareness of food quality and its impact on health, multidirectional research has been conducted, including on the content of bioactive compounds and potential factors affecting their content [1,2,3,4,5]. This includes endo- and exogenous factors that act on plants during vegetation and also after fruit harvest, shaping the fruit’s health-promoting potential [6,7]. Many studies are dedicated to the factors and conditions which should be ensured for the fruit to have the highest content of nutritionally valuable compounds after harvest and especially during storage [8]. Indeed, there is a significant relationship between the concentration of fruit health-promoting compounds and the species or cultivar [9,10,11], fruit part [5,12], or soil and climatic conditions during the growing season [11,13]. A separate issue in the context of fruit quality is harvesting and subsequent storage [14,15,16,17].

The storage potential of a fruit is an important characteristic for both the producer and the consumer. Fruits differ significantly in their storability, e.g., apples are characterized by a very high storability potential [6], whereas soft fruits retain their high quality in the unprocessed state for a relatively short time [4]. Research on the storage of the kiwifruit (*Actinidia deliciosa* (A. Chev.), C.F. Liang & A.R. Ferguson) in cold stores with low oxygen content (O_2_ = 0.5–1.0%) had already been conducted by the end of the 20th century [18]. Storing kiwifruit in 1% O_2_ is quite effective in inhibiting fruit ripening, but an oxygen content of less than 0.5% causes fermentation in the fruit. Developments in technology make it possible to reduce the oxygen content to a very low level, even to 0.2–0.4%. Studies indicate [17,19] that the practical use of low O_2_ levels is justified, as lowering the O_2_ content below 1.5% allows minikiwi (*Actinidia arguta* (Sieb. & Zucc.) Planch. ex Miq.) to be stored for more than 2 months under cold storage conditions equipped with dynamically controlled atmosphere (DCA) technologies. Reducing the O_2_ content to a limit just above the level of anaerobic respiration is effective in inhibiting the ripening of pome fruits such as apples [20], but this technology can also be used to store minikiwi. On the other hand, a study by Hertog et al. [21] showed that fruit stored in a low-carbon dioxide atmosphere, close to zero, had firmness about 15 N lower than fruit stored in an atmosphere with 5% CO_2_ concentration. DCA technology is based on measurements of ethanol release or chlorophyll fluorescence. Fruits reacting to oxygen stress switch from aerobic to anaerobic respiration, which can be identified, e.g., by monitoring chlorophyll fluorescence.

Minikiwi fruits are characterized by a high content of bioactive compounds such as polyphenols (including flavonoids), tannins, lutein, zeaxanthin, polysaccharides, and carotenoids (mainly β-carotene), as well as vitamins (vitamins B and C) and dietary fiber [10,22]. Many studies indicate that the presence of so many compounds with different beneficial mechanisms of action on the human body strengthens immunity and has a preventive effect against many diseases [9,23,24,25]. In addition, they are rich in essential minerals for the body such as copper, zinc, iron, potassium, and magnesium [26]. Polysaccharides are natural polymers that stimulate the human immune system. Polysaccharides from *Actinidia arguta* have been found to inhibit the oxidation of polyunsaturated fatty acids, which can cause the formation of lipid radicals that deteriorate food quality during storage [27]. The sugar that predominates in minikiwi fruit is sucrose, with glucose and fructose in lower concentrations [28].

When looking for optimum storage solutions for fruit, in addition to the easy-to-assess external quality, the question of its internal quality, i.e., maintaining the high nutritional value of the fruit, is no less important. The antioxidant content of the fruit is an important indicator of quality. The minikiwi is characterized by high vitamin C content [3,10]. Keeping the antioxidant content constant during fruit storage is an extremely important issue. During conventional cold storage, vitamin C content can change dynamically. Jeong et al. [29] reports that in fruits stored at 1 °C, the ascorbic acid content initially increases and then decreases dramatically. This process is much slower under ULO (Ultra Low Oxygen, 1.5% CO_2_ and 1.5% O_2_) conditions [4]. The antioxidant content determines the antioxidant activity of fruits. A study by Fisk et al. [30], however, showed no significant difference in total antioxidant activity (AA) during storage, either at room temperature (22 °C) or under refrigeration (2 °C). Furthermore, storage in CA (controlled atmosphere) has also often not resulted in large changes in AA of minikiwi fruit [4,29]. The total phenolic content of fruit changes during storage, but this process is related to the conditions under which the fruit is stored. Under room conditions (22 °C), a drastic reduction in phenolic content was observed [30], whereas lowering the temperature favors the inhibition of this process, although their content fluctuates during storage. Similar relationships were found for flavonoids; their contents also fluctuate frequently during storage at 1 °C, but remain fairly constant [29].

The aim of this study was to evaluate the influence of varying O_2_ and CO_2_ parameters, based on different storage technologies, on the nutritional value of minikiwi. The study attempted to demonstrate changes in the antioxidant potential occurring in fruit during storage due to very low-oxygen or high-carbon dioxide concentrations. The essence of the study was to determine the composition of an atmosphere with the lowest possible O_2_ concentration and the highest possible CO_2_ concentration, which effectively limits the loss of beneficial compounds in the minikiwi fruits.

## 2. Results

Fruits of both cultivars, immediately after harvest, were characterized by high antioxidant activity (AA) (Table 1). In the second year of the study, an increase in AA was observed, especially in ‘Geneva’. The high AA in this cultivar was probably determined by the high content of ascorbic acid and total polyphenols (TPC). In 2018, an increase in TPC, phenolic acids, and AA values was found in ‘Geneva’ fruit, while a decrease in ascorbic acid was unexpectedly observed in ‘Ananasnaya’ fruit. The ‘Geneva’ cultivar had a significantly higher postharvest sucrose content than the ‘Ananasnaya’ cultivar. The monosaccharide content was at a fairly similar level in ‘Geneva’ and ‘Ananasnaya’ fruits. The main acid found in the hardy kiwi is citric acid; the fruit contains amount five times less malic acid. In 2018, the acid content of the fruit of both cultivars was at a fairly similar level. In the second year of the study, a decrease in citric acid content was noted in ‘Geneva’ fruit, and similarly in malic acid content in ‘Ananasnaya’ fruit.

The effect of storage technology on the antioxidant potential of minikiwi was found in both years of the study (Table 2 and Table 3). The ‘Ananasnaya’ cultivar after storage in CA were characterized by significantly higher AA than fruits stored in other technologies. A similar relationship was found only in 2018 in fruit of the ‘Geneva’ cultivar. Higher AA in ‘Ananasnaya’ fruit stored in CA was found after 4 weeks in 2018 and 8 weeks in 2017, while in ‘Geneva’ fruit, it was only after 10 weeks in both years of the study (Appendix A). Also, ascorbic acid content was highly dependent on fruit storage conditions. In this case, the atmosphere with a higher CO_2_ content of 5% (CA) had a positive effect on the index value. Furthermore, TPC as well as other derivatives of the phenol group were clearly characterized by a higher content in fruits stored in CA. The effect of DCA technology was significantly weaker and the content of evaluated compounds in fruits of both cultivars was at a similar level as under ULO conditions. During storage, irrespective of the conditions applied, a constant decrease in AA in minikiwi as well as in the content of antioxidant compounds was observed. The significance of these changes depended largely on the cultivar and the index evaluated. The ‘Geneva’ cultivar fruits were characterized by a loss of antioxidant potential described by a decrease in the ascorbic acid and flavan-3-ol contents in both years of the study, while TPC, phenolic acids, and flavonols only decreased in 2017. In the ‘Ananasnaya’ cultivar fruit, significant changes after storage were recorded for all evaluated compounds in 2017 and for TPC and flavonols and flavan-3-ols derivatives in 2018. In both years of the study, a highly significant interaction was found between technology and storage period on the storage potential of minikiwi (Appendix A). In general, the reduction of AA and content of individual groups of evaluated compounds occurred significantly faster under ULO conditions and slowest under CA conditions. DCA technology contributed to a significant inhibition of potential loss but not as effectively as CA technology. In particular, the TPC content, including flavonols and flavan-3-ols derivatives, were stable under CA conditions. In this case, significantly lower contents of the mentioned compounds were observed after 4 weeks in fruits stored in ULO and DCA than in CA. In fruits of both cultivars, a lower phenolic acid derivative content as well as ascorbic acid content was found after 6 weeks of storage in ULO conditions compared to in CA. However, between DCA and CA, a difference in the content of the above mentioned compounds was observed after another 2 weeks (Appendix A).

Different concentrations of O_2_ and CO_2_, depending on the applied storage technology, did not affect the monosaccharide content (glucose and fructose) in the minikiwi fruit of both cultivars (Table 4 and Table 5). However, sucrose content changed significantly under storage conditions. An elevated CO_2_ concentration (CA) as well as a super-low O_2_ concentration (DCA) favored a higher content of this disaccharide. ULO technology proved to be the least beneficial in inhibiting sucrose breakdown over 12 weeks of storage. As expected, glucose and fructose contents increased significantly during fruit ripening in cold storage. After picking as well as after storage, the ‘Geneva’ cultivar fruit were characterized by higher contents of both monosaccharides than the ‘Ananasnaya’ cultivar fruits. Sucrose content decreased dramatically during storage, especially in the ‘Ananasnaya’ fruits, by 35% and 44% (in 2017 and 2018, respectively). Changes in monosaccharide content were determined by both factors of the experiment (Appendix A). Only ‘Ananasnaya’ fruits had higher monosaccharide content after storage in CA than in ULO, but only after 10 weeks in 2017 and 12 weeks in 2018. It was revealed that after storage of minikiwi in ULO, the content of glucose and fructose was significantly higher than after storage in CA. The monosaccharide content in fruits stored in DCA took intermediate values or was not statistically different from the monosaccharide content of minikiwi after storage in ULO. In contrast, sucrose content was more stable in minikiwi stored in CA or DCA, especially after 12 weeks. A significant loss of sucrose in fruits stored at ULO occurred in the first 4 weeks.

DCA—dynamic controlled atmosphere, 0.4% CO_2_:0.4% O_2_; ULO—ultra-low oxygen, 1.5% CO_2_:1.5% O_2_; CA—controlled atmosphere, 5% CO_2_:1.5% O_2_; *—interaction between storage conditions and storage time; data are presented as mean ± standard deviation.

The storage technology determined the content of citric and malic acids in minikiwi after storage (Table 6 and Table 7). The lowest content of both compounds was found in fruits stored under ULO cold storage conditions, while the highest acid content was found in fruits stored in CA. The significance of this interaction was not proven only in the case of citric acid in the ‘Geneva’ fruits. The storage period, irrespective of the technology, corresponded to a quite significant reduction in the content of both acids in the hardy kiwi. In this case, after 12 weeks of storage, the acid content decreased on average by about 35–40% in subsequent years of the study. Statistical analysis of the data showed a highly significant interaction of technology and storage period on the content of both evaluated acids in minikiwi (Appendix A). The results of the study indicate a differential effect of the technologies used during the study period. In both years of the study, CA technology was effective in reducing the reduction of citric acid in minikiwi. In contrast, the DCA technology in 2017 had low efficiency in this process, but already in 2018, its efficiency was much higher. Furthermore, this time, storage with the ULO technology proved to be the least effective in fixing the key quality characteristics of the hardy kiwi.

## 3. Discussion

The hardy kiwi (minikiwi) is a very interesting climber native to eastern Asia and Japan. Thanks to its high frost resistance, it can be cultivated in countries with temperate, continental climates characterized by low winter temperatures. For this reason, interest in the cultivation of minikiwi is growing worldwide, and the number of growers willing to establish commodity plantations is increasing from year to year.

Alongside the many advantages possessed by minikiwi, their disadvantage is their relatively short storage time and short fruit supply [4,17,31]. The storage potential of this species is closely related to their maturity at harvest [14,16]. Hardy kiwi fruit belongs to the climacteric fruit group and can be harvested before full consumer maturity. Such fruits are not suitable for consumption, but their storage period is extended [4,15,17]. A number of different post-harvest treatments, i.e., ethylene inhibitors [1], calcium chloride, or salicylic acid [32], have been tested to prolong the availability of minikiwi. An important factor in storage is the gas composition of the atmosphere in the cold storage. By controlling gas levels, it is possible to influence the rate of physiological processes occurring in fruit [17,19,33].

Changes in antioxidant compound content and their direction, occurring during storage are conditioned by the concentration of CO_2_ and O_2_ in the cold storage. Jeong et al. [29] and Szpadzik et al. [4] reported that in a normal atmosphere (NA, 0.1% CO_2_ and 21% O_2_), ascorbic acid content increases in the initial period and then decreases in the following storage days. Storage at 1 °C also causes a decrease in the total ascorbate content [1]. The results of our own study confirm the above relationship. The decrease in ascorbic acid content occurred throughout the storage period, but the rate of this process was determined by the storage technology. The atmosphere with a higher CO_2_ content of 5% (CA) effectively influenced the high value of the index even after 12 weeks of storage. In general, the decrease in ascorbic acid content was faster under ULO conditions and slowest under CA conditions. The oxidation of L-ascorbic acid to dehydroxyascorbate was responsible for the decrease in vitamin C content during longer storage.

The changes in antioxidant compound content described in the literature are not conclusive. Storage of fruit in a controlled atmosphere significantly reduces the decrease in phenolic content in fruit [4,29]. In an earlier study [30], a significant effect of storage temperature was shown, as the total phenolic content in fruits stored in cold storage at 2 °C was significantly higher than in fruits stored under room-temperature conditions (22 °C). Also according to Jeong et al. [29], fruits stored in cold storage have similar phenolic compound content compared to their post-harvest content. However, according to Fisk et al. [30] and Stefaniak et al. [1], the content of phenolic compounds in fruit stored in cold storge increased slightly, while other studies showed a decrease [33]. Similar relationships were found for flavonoid content, which was also either quite stable or slightly changed during low-temperature (1 °C) storage [4,32]. In our study, the direction of changes in the content of compounds from the phenolic group was dependent on the cultivar and storage conditions. The content of TPC decreased in both years of the study, while the content of other compounds fluctuated in both years of the study. However, it was shown that the rate of change was significantly dependent on storage conditions. In this case, DCA technology contributed to a significant inhibition of phenol loss but not as effectively as CA technology. AA is a measure of free radical scavenging capacity. The analyses showed that minikiwi, after storage in a CA, were characterized by a significantly higher AA than fruits stored in the other technologies. During storage, a steady reduction of AA in minikiwi was observed. An individually conducted experiment showed an effect of storage conditions on AA, which is consistent with reports by other authors highlighting the fact that AA is higher in less ripe fruit [21]. In contrast, Yildirima and Bayira [34] found that AA does not change significantly in fruit during storage.

Simple sugars are among the basic components providing energy to the human body, but polysaccharides contained within *A. arguta* inhibit the oxidation of polyunsaturated fatty acids in food [27]. The carbohydrate content of hardy kiwi fruit varies depending on the source cited; according to Latocha et al. [35], it is 13.6% on average. On the other hand, Park [36] give slightly lower values. Bieniek [13] reported that the fruits he studied contained from 6.00 to 7.54% carbohydrates. The carbohydrates found in the highest amounts in minikiwi fruits are fructose, glucose, and sucrose. In a study conducted by Wojdyło et al. [37] in minikiwi fruit depending on the cultivar found the following: 1.16–2.14 g·100 g^−1^ fructose, 1.57–2.99 g·100 g^−1^ F.W. glucose, and 2.41–5.94 g·100 g^−1^ F.W. sucrose. According to Park [36], the fruit also contains the alcoholic derivative of sugars myo-inositol at 6.0–10.46 µg·mg^−1^ F.W. This compound is very valuable for the human body, as it prevents cancer [38]. Barbonii et al. [39] found that the concentrations of simple sugars in minikiwi increase during cold storage, with more than a two-fold increase in the sugars assessed. In our own experiment, there was also a significant increase in glucose and fructose content, but a decrease in sucrose content. Glucose and fructose contents were found to be significantly higher after storage in ULO or DCA than after storage in CA. In contrast, sucrose content was more stable in fruit stored in CA or DCA. The beneficial effects of organic acids on human organisms have been known for many years; nevertheless, the knowledge of their mechanism of action is still being improved. Organic acids have an antibacterial effect; they are used in the treatment of digestive diseases and regulate blood glucose levels. Moreover, they play an important role in the assimilation of certain elements (e.g., calcium and iron) into the human body, as well as participating in the binding of heavy metals and their excretion [40,41]. These compounds are accumulated during fruit growth and are then used in respiration during ripening [42]. A study conducted by Barboni et al. [39] showed that citric acid, malic acid, and quinic acid contents decrease slightly during storage. The results of the experiment show the different influence of the technologies used during the study period. In both years of the study, the conditions of the CA technology stabilized the citric acid content of minikiwi. In contrast, the DCA technology in one year of the study was characterized by low efficiency in this process, but in the following year, its efficiency was much higher. In general, it can be concluded that elevated CO_2_ content is more effective in stabilizing the nutritional value of minikiwi during storage than a low-oxygen atmosphere.

## 4. Materials and Methods

### 4.1. Outline of the Experiment

The fruit originated from the experimental field of the Department of Pomology and Horticultural Economics, Warsaw University of Life Sciences (WULS-SGGW), located in Warsaw, central Poland (52.259° N, 21.020° E). The nutritional properties were assessed for two cultivars. One was ‘Geneva’, an early cultivar of minikiwi which is commonly grown in Poland [43]. Its characteristic feature is a completely green color of the skin and high fertility. The other was ‘Ananasnaya’, which is the basic cultivar grown in the United States and is the most widely grown minikiwi fruit in the world [44]. It is a variety with a late time of fruit ripening in the Polish climate. A characteristic blush appears on this fruit, but the skin retains an intensive green color. Both cultivars were selected for the study due to their high popularity in Poland, but different fruit ripening time. In terms of taste, it is considered to be one of the tastiest fruits manually harvested from 8-year-old *A. arguta* vines to plastic high-vented containers. During harvesting, the fruits of both cultivars were sorted, discarding fruits that differed in size (small) and had visible quality defects. The fruit was harvested at the harvest maturity phase, determined on the basis of the soluble solids content (SSC = 6–7° Brix), a method commonly used in commercial minikiwi plantations. This value has been suggested by other authors [4,45] as an appropriate value for the harvest of minikiwi for storage. Immediately after harvest, fruits were transported to the experimental storage of the Institute of Horticultural Sciences of Warsaw University of Life Sciences and stored in the experimental cold chamber (1 m^3^ capacity). The containers were equipped with an automatic system Oxystat 200 (David Bishoop Instruments, Heathfield E.Susex, United Kingdom), allowing continuous monitoring of CO_2_ and O_2_ levels, and Handy PEA fluorimeters (Hansatech Industries Ltd., Pentney, United Kingdom) to assess chlorophyll fluorescence. Instrument calibration was performed automatically with a calibration gas mixture every 48 h. The fruit was stored at 1 °C and about 90–95% relative air humidity. For the evaluation of the nutritional properties of the minikiwi, three gas mixtures corresponding to three storage technologies were used, i.e., controlled atmosphere, ultra-low oxygen, and dynamic controlled atmosphere. Under CA conditions, an air composition with 5% CO_2_:1.5% O_2_ was used. Under ULO conditions, the composition of the atmosphere was 1.5% CO_2_:1.5% O_2_. On the other hand, in DCA conditions, the composition of the atmosphere was dynamically maintained at a level of about 0.4% CO_2_ and about 0.4% O_2_, changing the oxygen content in periods of fruit stress by 0.1%. Fruit stress in DCA was identified by chlorophyll fluorescence. The combinations of air compositions used in this study were determined based on previous studies [4,19,46]. The stored fruit analyses were conducted 6 times, i.e., directly after harvest and at fourteen-day intervals during storage for up to 12 weeks of storage. The experiment was replicated three times, each on 0.5 kg of fruit (approx. 70–80 fruits). For analyses of bioactive compounds and antioxidant qualities, fruits were frozen in liquid nitrogen (quick freezing) and stored at −80 °C in preparation for the analysis.

### 4.2. Analytical Methods

All reagents were of analytical purity gradients or HPLC grade, purchased from Sigma-Aldrich (Poznan, Poland) or Merck (Warsaw, Poland).

Determination of vitamin C content (ascorbic acid content) was performed according to the HPLC method of Szpadzik et al. [47]. Ascorbic acid was extracted from a 50 g sample of fruit with the mixture of 3% (*w*/*v*) metaphosphoric (100 mL). Identification and quantitative analysis of AAC were performed using a series 200 HPLC system (Perkin Elmer, Krakow, Poland) equipped with a diode array detector (UV-DAD), using a Spheri-5 RP-18 column (5 µm, 220 mm × 4.6 mm, Brownlee Columns, Perkin Elmer, Krakow, Poland) at 1.0 mL/min flow-rate and detection at 245 nm. The mobile phase used water:ammonia phosphate:meta-phosphoric acid (98.75:0.25:1.0). The retention time was 2.12 min. Ascorbic acid content was identified on the basis of a standard and was expressed in mg·100 g^−1^ F.W. The total phenolic content (TPC) was determined by the spectrophotometric method [48], using the Folin–Ciocalteu reagent. The absorbance of the solution was measured using a Marcel 330S PRO spectrophotometer (Marcel, Zielonka, Poland) at the wavelength λ = 700 nm. Presented results were recalculated into gallic acid. Phenolic compounds were separated using the HPLC technique described in our previous studies [47]. Analysis of the separation and contents of phenolic compounds was performed using a Perkin-Elmer 200 series HPLC kit with a Diode Array Detector (DAD). Separation was carried out using a LiChroCART 125-3 (Merck KGaA, Darmstadt, Germany) column with a 1 mL/min flow rate. The column temperature was 22 °C. The mobile phase consisted of a water (A):20% formic acid (B):acetonitrile (C), with variable parameters of the gradients A and C; 0–20 min A:B:C = 63.5:22.5:14.0; 20–25 min A:B:C = 17.5:22.5:60.0; 25–35 min A:B:C = 75.5:22.5:2.0. Phenolic compounds were detected at 280, 300, 320, and 360 nm wavelengths by comparing retention time on achieved chromatograms with standard ones. Content of particular compounds (total of three groups: phenolic acids, flavonols, and flavan-3-ols) was given in mg·100 g^−1^ F.W. The antioxidant activity (AA) was determined according to the method by Saint Criq de Gaulejac et al. [49] based on the reduction of free radicals obtained from DPPH^+^ (1,1-diphenyl-2-picrylhydrazine, Sigma-Aldrich, Poznan, Poland). The AA was calculated on the basis of absorbance measurements for the proper sample (fruit extract + DPPH^+^) performed after 20 min at λ = 517 nm in relation to the control sample (H2O + DPPH^+^). The results were expressed in mg of ascorbic acid equivalent (AAE) per g of F.W. Sugars and organic acids were determined by HPLC-RI, as described previously by Zielinski et al. [50], and expressed as grams of total sugar content or organic acid per 100 g F.W.

### 4.3. Statistical Analysis

The results were analyzed statistically in Statistica 13.3 (StatSoft Polska, Krakow, Poland), using the two-way analysis of variance. Tukey test was used for evaluation of the significance of differences between the means, accepting the significance level as 5%.

## 5. Conclusions

The aim of the experiment was to determine the effect of low O_2_ concentration and high CO_2_ concentration on the nutritional value of fruits of Actinidia arguta (minikiwi). It was shown that the evaluated factors significantly determined fruit quality. Minikiwi, similarly to other berries, is not characterized by high storage ability and has very high antioxidant content, especially vitamin C. Research results indicate that the use of high CO_2_ concentration (5%) effectively inhibits the loss of polyphenols and antioxidant activity of the fruit. In ULO technology, this process occurs slightly faster. On the other hand, monosaccharide content is significantly higher after storage under DCA or ULO conditions. In general, high CO_2_ concentration had a stronger effect on the content of the evaluated antioxidants than super low O_2_ concentration, stabilizing their levels, which can be explained by the slower ripening of fruits under high CO_2_ conditions.

## Figures and Tables

**Table 1 molecules-27-04313-t001:** Characteristics of ‘Geneva’ and ‘Ananasnaya’ fruit assessed directly after harvest.

Cultivars	Geneva		Ananasnaya	
	2017	2018	2017	2018
AA (mg AAE·100 g^−1^ F.W.)	0.846 ± 0.06	1.151 ± 0.13	0.803 ± 0.07	0.939 ± 0.07
Ascorbic acid (mg·100 g^−1^ F.W.)	67.8 ± 6.5	70.1 ± 3.1	85.3 ± 8.6	70.2 ± 8.8
TPC (mg·100 g^−1^ F.W.)	110.1 ± 7.9	120.1 ± 1.1	101.9 ± 8.4	106.1 ± 8.5
Phenolic acids (mg·100 g^−1^ F.W.)	2.32 ± 0.15	3.39 ± 0.22	1.76 ± 0.06	1.59 ± 0.18
Flavonols (mg·100 g^−1^ F.W.)	1.66 ± 0.07	1.26 ± 0.04	6.58 ± 0.59	6.73 ± 0.78
Flavan-3-ols (mg·100 g^−1^ F.W.)	0.500 ± 0.02	0.496 ± 0.01	0.543 ± 0.04	0.624 ± 0.07
Glucose (g·100 g^−1^ F.W.)	1.91 ± 0.05	2.02 ± 0.08	1.55 ± 0.05	1.52 ± 0.03
Fructose (g·100 g^−1^ F.W.)	2.30 ± 0.04	2.53 ± 0.05	2.20 ± 0.07	1.98 ± 0.05
Sucrose (g·100 g^−1^ F.W.)	8.45 ± 0.31	8.00 ± 0.32	6.67 ± 0.04	6.64 ± 0.04
Citric acid (g·100 g^−1^ F.W.)	1.145 ± 0.09	0.871 ± 0.01	0.739 ± 0.02	0.884 ± 0.06
Malic acid (g·100 g^−1^ F.W.)	0.150 ± 0.02	0.131 ± 0.01	0.216 ± 0.03	0.127 ± 0.01

AA—antioxidant activity; TPC—total polyphenolic content; data are presented as mean ± standard deviation.

**Table 2 molecules-27-04313-t002:** Changes in antioxidant activity (mg vit C·100 g^−1^ F.W.) and ascorbic acid and polyphenol contents (mg·100 g^−1^ F.W.) measured in ‘Geneva’ and ‘Ananasnaya’ minikiwi fruit in the postharvest period in 2017.

Cultivar		Geneva					
		AA	Ascorbic acid	TPC	Phenolic acids	Flavonols	Flavan-3-ols
Storage conditions (A)	DCA	0.713 ± 0.10	57.8 ± 6.8	91.5 ± 10.5	1.90 ± 0.25	1.53 ± 0.13	0.416 ± 0.05
	ULO	0.706 ± 0.12	56.3 ± 8.6	89.0 ± 14.2	1.95 ± 0.27	1.51 ± 0.15	0.411 ± 0.06
	CA	0.761 ± 0.09	64.4 ± 6.1	109.8 ± 8.8	2.34 ± 0.14	1.76 ± 0.13	0.483 ± 0.04
	*p*-value	0.218	0.003	<0.001	<0.001	<0.001	<0.001
Time of storage (weeks) (B)	0	0.846 ± 0.06	67.8 ± 6.5	110.1 ± 7.9	2.32 ± 0.15	1.66 ± 0.07	0.500 ± 0.02
	4	0.746 ± 0.10	62.3 ± 7.0	98.6 ± 10.0	2.14 ± 0.17	1.65 ± 0.14	0.437 ± 0.04
	6	0.706 ± 0.06	58.1 ± 3.3	93.6 ± 13.0	2.03 ± 0.32	1.56 ± 0.23	0.422 ± 0.07
	8	0.717 ± 0.11	59.7 ± 9.5	97.7 ± 16.9	2.05 ± 0.29	1.63 ± 0.23	0.440 ± 0.06
	10	0.687 ± 0.07	55.6 ± 5.7	91.9 ± 14.4	1.95 ± 0.33	1.56 ± 0.17	0.413 ± 0.05
	12	0.657 ± 0.11	53.3 ± 6.5	88.7 ± 16.4	1.88 ± 0.32	1.54 ± 0.20	0.408 ± 0.06
	*p*-value	<0.001	<0.001	0.028	0.024	0.599	0.009
Interaction A*B	*p*-value	<0.001	<0.001	<0.001	<0.001	<0.001	<0.001
Cultivar		Ananasnaya					
		AA	Ascorbic acid	TPC	Phenolic acids	Flavonols	Flavan-3-ols
Storage conditions (A)	DCA	0.725 ± 0.08	71.7 ± 8.8	82.5 ± 12.3	1.47 ± 0.19	5.21 ± 0.84	0.451 ± 0.06
	ULO	0.677 ± 0.11	66.8 ± 11.8	83.7 ± 16.1	1.55 ± 0.23	5.72 ± 0.73	0.466 ± 0.07
	CA	0.816 ± 0.05	83.4 ± 7.6	99.4 ± 11.2	1.88 ± 0.16	6.44 ± 0.61	0.535 ± 0.05
	*p*-value	<0.001	<0.001	<0.001	<0.001	<0.001	<0.001
Time of storage (weeks) (B)	0	0.803 ± 0.07	85.3 ± 8.6	101.9 ± 8.4	1.76 ± 0.06	6.58 ± 0.59	0.543 ± 0.04
	4	0.781 ± 0.09	75.9 ± 8.3	91.2 ± 12.4	1.74 ± 0.20	5.90 ± 0.63	0.501 ± 0.05
	6	0.718 ± 0.08	72.6 ± 8.0	87.0 ± 13.7	1.59 ± 0.28	5.47 ± 0.76	0.477 ± 0.06
	8	0.743 ± 0.12	74.8 ± 13.6	93.3 ± 21.9	1.70 ± 0.34	6.02 ± 1.19	0.506 ± 0.11
	10	0.699 ± 0.10	69.1 ± 10.7	80.7 ± 9.4	1.57 ± 0.26	5.51 ± 0.71	0.443 ± 0.04
	12	0.693 ± 0.11	66.2 ± 12.5	77.2 ± 10.7	1.44 ± 0.27	5.27 ± 0.81	0.434 ± 0.06
	*p*-value	0.108	0.008	0.005	0.077	0.015	0.006
Interaction A*B	*p*-value	<0.001	<0.001	<0.001	<0.001	<0.001	<0.001

DCA—dynamic controlled atmosphere, 0.4% CO_2_:0.4% O_2_; ULO—ultra-low oxygen, 1.5% CO_2_:1.5% O_2_; CA—controlled atmosphere, 5% CO_2_:1.5% O_2_; AA—antioxidant activity; TPC—total polyphenolic content; * — interaction between storage conditions and storage time; data are presented as mean ± standard deviation.

**Table 3 molecules-27-04313-t003:** Changes in antioxidant activity (mg AAE·100 g^−1^ F.W.) and ascorbic acid and polyphenols contents (mg·100 g^−1^ F.W.) measured in ‘Geneva’ and ‘Ananasnaya’ minikiwi fruit in the postharvest period in 2018.

Cultivar		Geneva					
		AA	Ascorbic acid	TPC	Phenolic acids	Flavonols	Flavan-3-ols
Storage conditions (A)	DCA	1.011 ± 0.13	64.3 ± 5.6	102.8 ± 10.4	2.89 ± 0.38	1.15 ± 0.09	0.404 ± 0.06
	ULO	1.018 ± 0.15	62.5 ± 8.0	102.9 ± 13.2	3.07 ± 0.41	1.14 ± 0.11	0.415 ± 0.06
	CA	1.099 ± 0.13	71.7 ± 6.1	125.4 ± 9.8	3.60 ± 0.29	1.32 ± 0.09	0.485 ± 0.04
	*p*-value	0.121	<0.001	<0.001	<0.001	<0.001	<0.001
Time of storage (weeks) (B)	0	1.151 ± 0.13	70.1 ± 3.1	120.1 ± 1.1	3.39 ± 0.22	1.26 ± 0.04	0.496 ± 0.01
	4	1.086 ± 0.15	70.5 ± 7.9	113.6 ± 11.6	3.38 ± 0.35	1.25 ± 0.10	0.428 ± 0.04
	6	1.027 ± 0.09	65.7 ± 3.7	107.8 ± 15.0	3.10 ± 0.53	1.18 ± 0.13	0.425 ± 0.07
	8	1.074 ± 0.15	67.6 ± 10.7	112.9 ± 19.9	3.19 ± 0.55	1.23 ± 0.17	0.440 ± 0.07
	10	0.981 ± 0.10	63.0 ± 6.4	105.8 ± 15.9	3.13 ± 0.53	1.16 ± 0.12	0.421 ± 0.06
	12	0.937 ± 0.15	60.3 ± 7.4	102.1 ± 18.0	2.94 ± 0.50	1.14 ± 0.14	0.396 ± 0.07
	*p*-value	0.015	0.022	0.159	0.312	0.247	0.013
Interaction A*B	*p*-value	<0.001	<0.001	<0.001	<0.001	<0.001	<0.001
Cultivar		Ananasnaya					
		AA	Ascorbic acid	TPC	Phenolic acids	Flavonols	Flavan-3-ols
Storage conditions (A)	DCA	0.824 ± 0.08	61.4 ± 7.1	85.7 ± 13.1	1.49 ± 0.28	5.14 ± 0.99	0.512 ± 0.09
	ULO	0.770 ± 0.12	57.3 ± 9.3	91.8 ± 13.5	1.60 ± 0.22	5.70 ± 0.83	0.531 ± 0.09
	CA	0.923 ± 0.06	70.5 ± 7.5	107.5 ± 10.6	1.85 ± 0.27	6.38 ± 0.68	0.625 ± 0.06
	*p*-value	<0.001	<0.001	<0.001	<0.001	<0.001	<0.001
Time of storage (weeks) (B)	0	0.939 ± 0.07	70.2 ± 8.8	106.1 ± 8.5	1.59 ± 0.18	6.73 ± 0.78	0.624 ± 0.07
	4	0.853 ± 0.10	65.2 ± 10.2	96.0 ± 12.0	1.73 ± 0.30	5.71 ± 0.63	0.570 ± 0.05
	6	0.823 ± 0.06	61.0 ± 6.5	93.2 ± 14.4	1.58 ± 0.24	5.30 ± 0.94	0.539 ± 0.08
	8	0.829 ± 0.11	62.6 ± 9.5	99.7 ± 23.2	1.77 ± 0.33	5.96 ± 1.18	0.584 ± 0.12
	10	0.790 ± 0.13	59.0 ± 10.2	89.5 ± 10.6	1.66 ± 0.30	5.42 ± 0.78	0.517 ± 0.07
	12	0.800 ± 0.12	60.5 ± 10.3	85.6 ± 13.6	1.56 ± 0.39	5.33 ± 0.83	0.501 ± 0.10
	*p*-value	0.036	0.152	0.060	0.575	0.008	0.038
Interaction A*B	*p*-value	<0.001	<0.001	<0.001	<0.001	<0.001	<0.001

DCA—dynamic controlled atmosphere, 0.4% CO_2_:0.4% O_2_; ULO—ultra-low oxygen, 1.5% CO_2_:1.5% O_2_; CA—controlled atmosphere, 5% CO_2_:1.5% O2; AA—antioxidant activity; TPC—total polyphenolic content; *—interaction between storage conditions and storage time; data are presented as mean ± standard deviation.

**Table 4 molecules-27-04313-t004:** Changes in monosaccharide and sucrose contents (g·100 g^−1^ F.W.) measured in ‘Geneva’ and ‘Ananasnaya’ minikiwi fruit in the postharvest period in 2017.

Cultivars		Geneva			Ananasnaya		
		Glucose	Fructose	Sucrose	Glucose	Fructose	Sucrose
Storage conditions (A)	DCA	2.99 ± 0.59	3.19 ± 0.50	7.29 ± 0.65	2.39 ± 0.48	3.14 ± 0.50	4.57 ± 1.07
	ULO	3.16 ± 0.65	3.46 ± 0.67	6.65 ± 0.99	2.42 ± 0.42	3.18 ± 0.48	3.97 ± 1.34
	CA	3.01 ± 0.62	3.17 ± 0.50	7.63 ± 0.58	2.50 ± 0.56	3.32 ± 0.69	5.22 ± 0.83
	*p*-value	0.670	0.234	0.0012	0.809	0.579	0.006
Time of storage (weeks) (B)	0	1.91 ± 0.05	2.30 ± 0.04	8.45 ± 0.31	1.55 ± 0.05	2.20 ± 0.07	6.67 ± 0.04
	4	2.73 ± 0.16	2.93 ± 0.12	7.01 ± 0.52	2.17 ± 0.15	2.98 ± 0.16	4.83 ± 0.66
	6	3.08 ± 0.17	3.19 ± 0.17	6.28 ± 0.58	2.49 ± 0.14	3.28 ± 0.20	4.01 ± 0.47
	8	3.59 ± 0.16	3.67 ± 0.22	7.43 ± 0.43	2.81 ± 0.07	3.61 ± 0.09	4.28 ± 0.94
	10	3.54 ± 0.06	3.71 ± 0.19	7.14 ± 0.59	2.82 ± 0.15	3.64 ± 0.23	4.00 ± 0.71
	12	3.47 ± 0.18	3.83 ± 0.27	6.82 ± 0.75	2.78 ± 0.26	3.58 ± 0.43	3.73 ± 0.86
	*p*-value	<0.001	<0.001	<0.001	<0.001	<0.001	<0.001
Interaction A*B	*p*-value	<0.001	<0.001	<0.001	<0.001	<0.001	<0.001

DCA—dynamic controlled atmosphere, 0.4% CO_2_:0.4% O_2_; ULO—ultra-low oxygen, 1.5% CO_2_:1.5% O_2_; CA—controlled atmosphere, 5% CO_2_:1.5% O_2_; *—interaction between storage conditions and storage time; data are presented as mean ± standard deviation.

**Table 5 molecules-27-04313-t005:** Changes in monosaccharide and sucrose contents (g·100 g^−1^ F.W.) measured in ‘Geneva’ and ‘Ananasnaya’ minikiwi fruit in the postharvest period in 2018.

Cultivars		Geneva			Ananasnaya		
		Glucose	Fructose	Sucrose	Glucose	Fructose	Sucrose
Storage conditions (A)	DCA	3.01 ± 0.59	3.56 ± 0.59	6.92 ± 0.58	2.31 ± 0.50	2.79 ± 0.47	5.10 ± 0.83
	ULO	3.19 ± 0.69	3.85 ± 0.77	6.53 ± 0.79	2.45 ± 0.52	2.94 ± 0.51	4.63 ± 1.03
	CA	2.96 ± 0.61	3.53 ± 0.60	7.16 ± 0.51	2.38 ± 0.56	2.87 ± 0.57	5.59 ± 0.61
	*p*-value	0.510	0.296	0.0163	0.758	0.697	0.005
Time of storage (weeks) (B)	0	2.02 ± 0.08	2.53 ± 0.05	8.00 ± 0.32	1.52 ± 0.03	1.98 ± 0.05	6.64 ± 0.04
	4	2.59 ± 0.16	3.23 ± 0.13	7.00 ± 0.36	1.97 ± 0.14	2.54 ± 0.12	5.51 ± 0.46
	6	2.92 ± 0.18	3.54 ± 0.19	6.54 ± 0.33	2.28 ± 0.13	2.79 ± 0.16	4.79 ± 0.46
	8	3.53 ± 0.15	4.12 ± 0.22	6.71 ± 0.31	2.78 ± 0.07	3.25 ± 0.13	4.85 ± 0.59
	10	3.55 ± 0.17	4.16 ± 0.21	6.41 ± 0.42	2.79 ± 0.08	3.27 ± 0.09	4.57 ± 0.52
	12	3.70 ± 0.16	4.31 ± 0.27	6.11 ± 0.45	2.93 ± 0.11	3.37 ± 0.15	4.29 ± 0.63
	*p*-value	<0.001	<0.001	<0.0001	<0.001	<0.001	<0.001
Interaction A*B	*p*-value	<0.001	<0.001	<0.0001	<0.001	<0.001	<0.001

DCA—dynamic controlled atmosphere, 0.4% CO_2_:0.4% O_2_; ULO—ultra-low oxygen, 1.5% CO_2_:1.5% O_2_; CA—controlled atmosphere, 5% CO_2_:1.5% O_2_; *—interaction between storage conditions and storage time; data are presented as mean ± standard deviation.

**Table 6 molecules-27-04313-t006:** Changes in citric acid and malic acid contents (g·100 g^−1^ F.W.) measured in ‘Geneva’ and ‘Ananasnaya’ minikiwi fruit in the postharvest period in 2017.

Cultivars		Geneva		Ananasnaya	
		Citric acid	Malic acid	Citric acid	Malic acid
Storage conditions (A)	DCA	0.834 ± 0.19	0.112 ± 0.03	0.593 ± 0.09	0.172 ± 0.03
	ULO	0.791 ± 0.20	0.108 ± 0.03	0.528 ± 0.13	0.155 ± 0.04
	CA	0.875 ± 0.15	0.126 ± 0.02	0.636 ± 0.07	0.212 ± 0.04
	*p*-value	0.389	0.068	0.009	<0.001
Time of storage(weeks) (B)	0	1.145 ± 0.09	0.150 ± 0.02	0.739 ± 0.02	0.216 ± 0.03
	4	0.885 ± 0.10	0.120 ± 0.01	0.645 ± 0.05	0.205 ± 0.03
	6	0.791 ± 0.10	0.119 ± 0.02	0.596 ± 0.06	0.197 ± 0.05
	8	0.790 ± 0.07	0.111 ± 0.01	0.555 ± 0.06	0.150 ± 0.03
	10	0.728 ± 0.12	0.101 ± 0.02	0.522 ± 0.07	0.168 ± 0.03
	12	0.660 ± 0.09	0.090 ± 0.02	0.457 ± 0.10	0.143 ± 0.04
	*p*-value	<0.001	<0.001	<0.001	<0.001
Interaction A*B	*p*-value	<0.001	<0.001	<0.001	<0.001

DCA—dynamic controlled atmosphere, 0.4% CO_2_:0.4% O_2_; ULO—ultra-low oxygen, 1.5% CO_2_:1.5% O_2_; CA—controlled atmosphere, 5% CO_2_:1.5% O_2_; *—interaction between storage conditions and storage time; data are presented as mean ± standard deviation.

**Table 7 molecules-27-04313-t007:** Changes in citric acid and malic acid contents (g·100 g^−1^ F.W.) measured in ‘Geneva’ and ‘Ananasnaya’ minikiwi fruit in the postharvest period in 2018.

Cultivars		Geneva		Ananasnaya	
		Citric acid	Malic acid	Citric acid	Malic acid
Storage conditions (A)	DCA	0.681 ± 0.11	0.108 ± 0.02	0.693 ± 0.12	0.098 ± 0.02
	ULO	0.659 ± 0.12	0.097 ± 0.02	0.623 ± 0.17	0.084 ± 0.03
	CA	0.729 ± 0.09	0.116 ± 0.02	0.765 ± 0.09	0.112 ± 0.02
	*p*-value	0.139	0.017	0.006	0.001
Time of storage (weeks) (B)	0	0.871 ± 0.01	0.131 ± 0.01	0.884 ± 0.06	0.127 ± 0.01
	4	0.739 ± 0.05	0.113 ± 0.01	0.743 ± 0.04	0.115 ± 0.01
	6	0.697 ± 0.04	0.114 ± 0.01	0.715 ± 0.08	0.107 ± 0.02
	8	0.652 ± 0.04	0.107 ± 0.01	0.659 ± 0.09	0.084 ± 0.01
	10	0.615 ± 0.04	0.096 ± 0.02	0.614 ± 0.12	0.079 ± 0.02
	12	0.565 ± 0.07	0.080 ± 0.02	0.549 ± 0.12	0.076 ± 0.02
	*p*-value	<0.001	<0.001	<0.001	<0.001
Interaction A*B	*p*-value	<0.001	<0.001	<0.001	<0.001

DCA—dynamic controlled atmosphere, 0.4% CO_2_:0.4% O_2_; ULO—ultra-low oxygen, 1.5% CO_2_:1.5% O_2_; CA—controlled atmosphere, 5% CO_2_:1.5% O_2_; *—interaction between storage conditions and storage time; data are presented as mean ± standard deviation.

## Data Availability

Not applicable.

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
