# Peer review of "Nutritional Values of Minikiwi Fruit (Actinidia arguta) after Storage: Comparison between DCA New Technology and ULO and CA"

_molecules, 2022, doi:10.3390/molecules27134313_

Round 1

Reviewer 1 Report

The work “Nutritional values of Minikiwi fruit (Actinidia arguta) after storage: comparison between DCA new technology and ULO and CA” by Krupa et al is in my opinion a very relevant, was carefully planned and executed and I recommend its publication after minor reviews and considerations that in my view should be discussed in the MS.

          The authors studied the variation of several parameters along 12 weeks of fruits storage and report that the nutritional value of the fruit after storage in CA or DCA was not significantly reduced, which will allow the supply of fresh minikiwi fruit to be extended and provide a valuable component of the human diet. However, there seem to be specific variations in the glucose and fructose contents (be significantly higher after storage in ULO or DCA, while sucrose content was more stable in fruit stored in CA or DCA) as well as in the citric acid content (CA technology conditions stabilized it, while DCA technology was less effective in inhibiting acid loss) which certainly will affect the fruit flavour and taste. How significant are those variations in the fruit flavour and taste throughout the 12 weeks of the study? In the end, despite the nutritional value being maintained, the fruit flavour and taste will be the most important criteria for the consumer to accept or reject the stored fruit. Did the authors verify this aspect? A sensory panellist of consumers or trained personnel was considered in this study? And how this eventual deterioration in flavour and taste is dependent on the cultivar?

-          Regarding the two cultivars of minikiwi fruit used in this study, few information is provided. Instead, the authors refer to references 43 and 44. In my view, it would be important to highlight some differences between both cultivars because it seems to me that, despite the study being performed with 8 years old vines, the Geneva cultivar seems to be better adapted to the continental climate of Poland than the Ananasnaya coming from the USA. Is this correct, of both cultivars grown equally well in Poland?

Minor points  

-       There are several typos throughout the MS that authors should correct, namely the O2 and CO2 numbers format that should be in lower script, as well as other lower capital letters in the abstract (line 16), a dot is missing (line 80), etc

-       DCA and ULO abbreviations should be explained also in the abstract

Author Response

Dear Reviewer,

We wanted to thank you for your thorough review of the manuscript "Nutritional values of Minikiwi fruit (Actinidia arguta (Sieb. & Zucc.) Planch. ex Miq.) after storage: comparison between DCA new technology and ULO and CA".

The comments made in the review are very valid and pertinent. We shall endeavor use them in the preparation of another article on the storage of minikiwi and the influence of the applied technology on the widely understood storage quality of hardy kiwi fruit.

  1. Unfortunately, in the experiment conducted and described in the manuscript, the authors did not carry out either a sensory or consumer evaluation of the minikiwi during the experiment. It is therefore not possible to supplement the article with this information. In another experiment on the storage of minikiwi fruit under conventional, controlled atmosphere and modified atmosphere cold storage conditions, sensory evaluation studies (consumer panel) were conducted. The results of this experiment are being prepared for publication, but due to the different storage conditions and different year of the study cannot be included in this manuscript.
  2. ‘Geneva’ cultivar was selected at a research station in Geneva, New York. It is a variety with an early time of ripening under the conditions of Polish climate. Its characteristic feature is completely green colour of the skin and high fertility. In terms of taste it is considered to be one of the tastiest. ‘Ananasnaya’ cultivar is of unknown origin. It gained very high recognition in the USA, therefore it is widely cultivated in North America, but also in other countries, such as Poland. It is a variety with a late time of fruit ripening in the Polish climate. On its fruits a characteristic blush appears, but the skin retains an intensive green colour. Both cultivars were selected for the study due to their high popularity in Poland, but different fruit ripening time. Both varieties grow equally well in Poland. The authors supplemented the information on both varieties in the manuscript.
  3. The manuscript has been corrected for any errors noted - lower and upper indices, missing letters or other editing errors. The text has been proofread by a native speaker.

The authors hope that the above explanations and adherence to the suggestions made in the reviews will render the attached manuscript appropriate and free from any understatement.

Yours sincerely

Tomasz Krupa Ph.D.

Department of Pomology and Horticultural Economics

Warsaw University of Life Science, Poland.

Reviewer 2 Report

comments are in document

Author Response

Dear Reviewer,

We wanted to thank you for your thorough and detailed review of the manuscript "Nutritional values of Minikiwi fruit (Actinidia arguta (Sieb. & Zucc.) Planch. ex Miq.) after storage: comparison between DCA new technology and ULO and CA". The comments in the review are very valid and pertinent. Efforts have been made to improve the manuscript taking into account the reviewer's comments.

  1. Comments 1; 3; 8; 11; 21 and 22

Any comments on editing errors (missing indexes; misplaced explanations of abbreviations; missing letters (citric acid, etc.); explanation of abbreviation F.W.; DPPH+) have been corrected in the manuscript.

  1. Comment 2

Efforts were made to correct language errors and the text was checked by a native speaker.

  1. Comments 4 and 5

Comments on the Abstract and keywords have been corrected.

  1. Comments 6

The change in oxygen content of DCA (plus or minus 0.1%) was based on the measurement of chlorophyll fluorescence. Fluorescence excitation occurs as a result of putting the fruit under stress with too low an oxygen concentration - anaerobic respiration. An increase of 0.1% in the oxygen concentration in the cold store terminates the stress of the fruit - a return to aerobic respiration, resulting in a decrease in chlorophyll fluorescence. In this situation the oxygen concentration is again reduced by 0.1%. Therefore the target oxygen concentration in the DCA was 0.4%. Every several days the oxygen concentration was increased to 0.5% (fluorescence increase) and after 24 hours it was decreased to 0.4% (fluorescence decrease).

  1. Comment 7

As required by the journal Molecules, the Materials and Methods chapter should follow the Discussion chapter. The authors have no influence on the established layout of the manuscript and cannot interfere or make changes.

  1. Comment 9

The authors have re-examined the description of the results, making changes in the sections that need improvement.

  1. Comment 10

The phrase "super-low O2 concentration" was used to distinguish DCA technology (abbreviation given in the text is DCA) from ULO - ultra-low oxygen technology.

  1. Comment 12

The authors have prepared tables in which the data are presented as simple factors rather than interactions between the effects of technology and storage period. This presentation of results is clearer and often used in the literature. Adding the results of the interaction between technology and storage period to the tables causes a lack of legibility of the presented data - too many data. Therefore, in order to illustrate the impact of individual technologies over time, it was decided to include interaction tables in the appendices to the manuscript.

  1. Comments 13 - 16

The reviewer's comments are accurate. When preparing the manuscript, the authors decided to add a PCA analysis to demonstrate the relationship. However, after supplementing the manuscript with additional tables showing the interaction of technology and storage period (in the appendices), the authors decided to remove the PCA analysis from the manuscript, as suggested by the reviewer. The authors agree that Anova is an appropriate and sufficient statistical tool to demonstrate the relationship between technology and storage time on selected distinguishing characteristics of minikiwi nutritional properties.

  1. Comments 17 and 18

Redundant or duplicate information in the text was removed

  1. Comment 19

A method for ascorbic acid determination was developed over 10 years ago (Krupa T., Latocha P., Liwińska A. 2011. Changes of physicochemical quality, phenolics and vitamin C content in hardy kiwifruit. Sci. Hort.130 (20): 410-417. https://doi.org/10.1016/j.scienta.2011.06.044), based on the literature of the time and the Polish Standard PN-EN ISO/IEC 17025. Ascorbic acid was eluted with a mixture of water : ammonium phosphate : meta-phosphoric acid (98.75:0.25:1.00). A simplification used erroneously by the authors was to state that ascorbic acid was eluted in 1% phosphoric acid. The text of the manuscript has been corrected and retention time information has been added.

  1. Comment 20

The method for the determination of polyphenolic compounds using HPLC was also developed on the basis of literature more than 10 years ago (Krupa T., Latocha P., Liwińska A. 2011. Changes of physicochemical quality, phenolics and vitamin C content in hardy kiwifruit. Sci. Hort.130 (20): 410-417. https://doi.org/10.1016/j.scienta.2011.06.044). The fragment concerning the description of the method was supplemented with missing information indicated by the reviewer.

The authors hope that the above explanations and adherence to the suggestions made in the reviews will render the attached manuscript appropriate and free from any understatement.

Kind regards,

Tomasz Krupa Ph.D.

Department of Pomology and Horticultural Economics

Warsaw University of Life Science, Poland.